# Sevoflurane Exposure in Neonates Perturbs the Expression Patterns of Specific Genes That May Underly the Observed Learning and Memory Deficits

**DOI:** 10.3390/ijms24108696

**Published:** 2023-05-12

**Authors:** Nerea Jimenez-Tellez, Marcus Pehar, Frank Visser, Alberto Casas-Ortiz, Tiffany Rice, Naweed I. Syed

**Affiliations:** 1Department of Biochemistry and Molecular Biology, University of Calgary, Calgary, AB T2N 4N1, Canada; nerea.jimeneztell1@ucalgary.ca (N.J.-T.); alberto.casasortiz@ucalgary.ca (A.C.-O.); 2Hotchkiss Brain Institute, University of Calgary, Calgary, AB T2N 4N1, Canada; marcus.pehar1@ucalgary.ca (M.P.); fvisser@ucalgary.ca (F.V.); 3Alberta Children’s Hospital Research Institute, University of Calgary, Calgary, AB T2N 4N1, Canada; tiffany.rice@albertahealthservices.ca; 4Department of Anesthesiology, Perioperative and Pain Medicine, University of Calgary, Calgary, AB T2N 4N1, Canada; 5Department of Cell Biology and Anatomy, University of Calgary, Calgary, AB T2N 4N1, Canada

**Keywords:** sevoflurane, dexmedetomidine, anesthetics, learning and memory, mitochondria

## Abstract

Exposure to commonly used anesthetics leads to neurotoxic effects in animal models—ranging from cell death to learning and memory deficits. These neurotoxic effects invoke a variety of molecular pathways, exerting either immediate or long-term effects at the cellular and behavioural levels. However, little is known about the gene expression changes following early neonatal exposure to these anesthetic agents. We report here on the effects of sevoflurane, a commonly used inhalational anesthetic, on learning and memory and identify a key set of genes that may likely be involved in the observed behavioural deficits. Specifically, we demonstrate that sevoflurane exposure in postnatal day 7 (P7) rat pups results in subtle, but distinct, memory deficits in the adult animals that have not been reported previously. Interestingly, when given intraperitoneally, pre-treatment with dexmedetomidine (DEX) could only prevent sevoflurane-induced anxiety in open field testing. To identify genes that may have been altered in the neonatal rats after sevoflurane and DEX exposure, specifically those impacting cellular viability, learning, and memory, we conducted an extensive Nanostring study examining over 770 genes. We found differential changes in the gene expression levels after exposure to both agents. A number of the perturbed genes found in this study have previously been implicated in synaptic transmission, plasticity, neurogenesis, apoptosis, myelination, and learning and memory. Our data thus demonstrate that subtle, albeit long-term, changes observed in an adult animal’s learning and memory after neonatal anesthetic exposure may likely involve perturbation of specific gene expression patterns.

## 1. Introduction

Anesthetics are essential for animal and human surgeries for pain management [1], dentistry practices [2], electroconvulsive therapy [3], or depression treatment [4]. Various anesthetic compounds have evolved over the years to enable improved clinical practices and care [5,6]. However, despite the development of novel anesthetic compounds and improvements in the practice of anesthesia, several animal studies, including those on primates, continue to raise questions about their short- and long-term cytotoxic effects [7]. Numerous human studies, albeit inconclusive, have also raised concerns, directing the Food and Drug Administration (FDA) to recommend warning labels on anesthetics—specifically when applied to pregnant women and pediatric patients [8]. However, the precise mechanisms underlying the long-term effects of anesthetic-induced toxicity have not been fully elucidated at the cellular, behavioural and gene expression levels.

One of the most commonly used anesthetics in pediatric patients is sevoflurane due to its lower irritability and faster onset and offset properties [9] compared to other agents. A few studies have demonstrated its cytotoxic effects on the developing brain in rodent models [10,11,12,13] and non-human primates [14,15], while the evidence for any adverse effects of sevoflurane in humans remains equivocal [16,17,18,19]. It is also important to note that the evidence regarding the anesthetic-induced detrimental effects observed in various animal models does not often reconcile with clinical practices [16,17,18,19] because neither the concentrations nor the exposure times of sevoflurane have been kept consistent [20,21,22,23]. Moreover, whereas the anesthetic-induced cytotoxic effects and the impact of long-term exposure in animal models have been documented, the precise mechanisms involving these detrimental effects remain unknown.

It is important to note that while anesthetic-induced cytotoxic effects have previously been observed at the cellular, molecular [12,24,25,26,27], behavioural [21,27,28,29,30], and gene expression levels [31,32,33,34], no single study has provided either a direct correlation or demonstrated interdependence between the observed changes. Moreover, there does not exist consensus in the literature regarding the involvement of any given pathway mediating anesthetic-induced toxicity. For instance, two different studies have claimed that early exposure to sevoflurane in rats, either once [31] or three times [32], results in the downregulation of Krüppel-like factor 4 (Kfl4), a transcription factor involved in cell growth, proliferation, or differentiation [35], over both a shorter [31] and a longer term [32]. In contrast, other studies have shown completely different subsets of genes to be altered following anesthetic exposure [33,34,36,37,38,39,40]. However, in these studies, neither the developmental stages nor the exposure conditions were considered for a fair comparison [33,34,36,37,38,39,40], thus making these findings inconclusive. Other studies using different anesthetics such as isoflurane [41,42,43,44], propofol [34,43] or ketamine [43,45] have also failed to provide a consensus for specific genes altered by anesthetic exposure, as neither the models used nor the experimental conditions were kept consistent. Moreover, it has also not been possible to separate the observed changes in gene expression resulting from surgical procedures and/or the ensuing pain from the anesthetic-induced effects on gene expression [42].

Recent studies have focused on a newer anesthetic adjunct, dexmedetomidine (DEX), which differs from other commonly used agents, as its mechanism of action relies upon targeting α2 adrenergic receptors [46] instead of Gamma-aminobutyric acid (GABA) receptors [47,48] or N-methyl-D-aspartate (NMDA) receptors [49,50] that are implicated in learning and memory [7]. Different animal studies have focused on DEX use as a co-adjuvant along with other commonly used anesthetics to assess its possible neuroprotective effects. These studies have shown that DEX-mediated neuroprotective effects may involve suppression of apoptotic pathways [13,51,52], inflammation [53,54], or neurogenesis [12,55]. However, an extensive analysis at the level of gene expression has not been performed to better understand how DEX may override the neurotoxic effects of sevoflurane.

In this study, firstly we examined the effects of clinically relevant concentrations and exposure times of sevoflurane in neonatal rats on learning and memory and the impact of DEX pre-treatment on those outcomes. Secondly, we sought to identify genes that could potentially be involved in the induced long-term learning and memory deficits.

## 2. Results

### 2.1. Sevoflurane Exposure Did Not Compromise Locomotor Skills but Impacted Various Aspects of Exploratory Behaviours

To determine if neonatal rat exposure to clinically relevant concentrations of sevoflurane resulted in long-term detrimental effects, as we had previously reported with subclinical concentrations [27], we exposed P7 rats to sevoflurane and subsequently examined levels of anxiety, locomotor skills, and “willingness” to explore a new environment during open field testing as previously described [27]. To analyze these aspects of animal learning and memory, the box where the assay took place was divided into four areas: corners, walls, inner areas, and centre, using ANY-Maze. The parameters analyzed were mobile time (Figure 1a), immobile time (Figure 1b), immobile ratio (Figure 1e) and total distance travelled within the box (Figure 1d), where averages per area were compared between the groups using one-way ANOVA statistical testing. We found no significant differences in any of the above four parameters between various groups. The other two parameters analyzed were time spent (Figure 1c) and the number of entries made to each area, respectively (Figure 1e). In all the groups, the preferences were: corners > walls > inner areas > centre, which were indicative of a consistent pattern of “anxiety”-like behaviour where the animals preferentially stayed in the most secluded areas. Interestingly, there was a significant increase in the time spent in the corners by the animals that were exposed to sevoflurane for 2 h (mean = 549.9, SEM = 31.75, *p* = 0.0116, *n* = 7) compared to their control (saline only) counterparts (mean = 458.8, SEM = 17.42, *n* = 9).

Similarly, we documented the preference for each area between the groups using heatmaps (Figure 1g–i) and discovered that the animals exposed to sevoflurane for 2 h had a strong preference for the bottom right corner (Figure 1i), whereas the animals exposed for only 1 h preferentially moved around the top right and bottom left corners (Figure 1h), in complete contrast to the control animals who explored all corners equally except for the top left one (Figure 1g).

These data demonstrate that clinically relevant doses of sevoflurane did not impact locomotor skills, as all exposed animals spent similar time exploring the new environment. However, with longer exposure times to sevoflurane, the exposed animals appeared ti exhibit little to no exploratory phenotype.

### 2.2. Sevoflurane Exposure Resulted in Erratic Swimming Patterns but Did Not Impact Spatial Memory

To assess whether a clinically relevant concentration of sevoflurane had any long-term effects on spatial learning, we used Morris water maze (MWM) as previously described [27,55]. Latency (time required to reach the hidden platform), distance swum, and the trajectory followed by the animals while swimming was monitored during each trial. Two-way ANOVA statistical analysis was performed to compare the average latency and distance between days 1 to 5 for each of the groups. For the latency, there were no significant differences between the sevoflurane groups, as the animals significantly decreased the time required to reach the platform between days 1 and 5 (sevoflurane 3.2% 1 h (day 1: mean = 44.97, SEM = 2.82, *n* = 9 and day 5: mean = 19.11, SEM = 3.638, *p* < 0.0001, *n* = 9) and sevoflurane 3.2% 2 h (day 1: mean = 44.03, SEM = 3.704, *n* = 9 and day 5: mean = 17.86, SEM = 3.525, *p* < 0.0001, *n* = 9)), as did their control counterparts (day 1: mean = 45.6, SEM = 1.986, *n* = 28 and day 5: mean = 20.75, SEM = 1.74, *p* < 0.0001, *n* = 28), indicating that spatial learning did indeed occur in all of the groups.

For the distance swum, there was no significant improvement between days 1 and 5 in the animals exposed to sevoflurane (Figure 1k) (sevoflurane 3.2% 1 h (day 1: mean = 12.98, SEM = 1.572, *n* = 9 and day 5: mean = 4.244, SEM = 0.9956, *p* = 0.754, *n* = 9) and sevoflurane 3.2% 2 h (day 1: mean = 15.22, SEM = 1.846, *n* = 9 and day 5: mean = 4.517, SEM = 1.25, *p* = 0.42, *n* = 9)), in contrast to their control counterparts (day 1: mean = 20.14, SEM = 3.356, *n* = 28 and day 5: mean = 6.328, SEM = 1.191, *p* < 0.0001, *n* = 28). These data were consistent with the swimming patterns on day 5 (Figure 1n–q). Notably, when we examined the trajectories followed in quadrants 1 (Figure 1n) and 3 (Figure 1p), we observed that the animals exposed to sevoflurane exhibited more erratic behaviour and could not locate the platform easily; they took considerably larger swimming routes to reach the platform, indicating that they did not exhibit significant improvement for the distance swum (Figure 1k).

To further understand the long-term impact on learning and memory, we next tested the effects of sevoflurane exposure on memory retention. For this, the animals were subjected to repeat MWM testing over time, on days 14, 28, and 56, respectively. During the non-testing time that lapsed in between, the animals were not allowed to go anywhere near the pool. We analyzed latency (Figure 1l) and distance swum (Figure 1m) for this group. We did not observe any significant difference in either of these parameters for any of the groups, suggesting that the learning acquired during the initial 5-day phase persisted over time and that sevoflurane did not affect memory consolidation.

### 2.3. Sevoflurane Exposure Resulted in Object Recognition Deficits

To test for the hippocampal-dependent recognition memory, we used the novel object recognition test (NORT) where the discrimination index ((DI = TN/(TN + TF), where TN = time spent while exploring the new object or the object placed at a different location, and TF = time spent exploring the familiar object) and the relative time exploring the object were measured. Interestingly, the animals exposed to sevoflurane did not exhibit a significant increase in the time spent with the novel object in the testing phase (Sev 3.2% 1 h: mean = 0.6085, SEM = 0.0449, *p* = 0.5681, *n* = 9 and Sev 3.2% 2 h: mean = 0.5782, SEM = 0.0829, *p* > 0.9999, *n* = 9) as compared with the familiarization phase (Sev 3.2% 1 h: mean = 0.4962, SEM = 0.0359, *n* = 9 and Sev 3.2% 2 h: mean = 0.5769, SEM = 0.0281, *n* = 9), in contrast to what we had observed in the control group (testing phase: saline: mean = 0.7135, SEM = 0.0422, *p* = 0.0031, *n* = 11; familiarization phase: saline: mean = 0.4706, SEM = 0.0291, *n* = 11) (Figure 2a). Intriguingly, with the longer exposure time (Sev 3.2% 2 h), both familiarization and testing phases were almost identical, with the animals not being aware of the novelty of the introduced object. With respect to the relative time spent interacting with both objects during the phases, there were no significant differences between the groups, ruling out that the overall exploration time could have been responsible for the differences observed during the introduction of the novel object.

With regard to the change involving the location of one of the familiar objects, in all groups the discrimination index was statistically significant, being higher in the testing phase (Sev 3.2% 1 h: mean = 0.6217, SEM = 0.0408, *p* = 0.0030, *n* = 9 and Sev 3.2% 2 h: mean = 0.682, SEM = 0.0413, *p* = 0.0028, *n* = 9) compared to the familiarization phase (Sev 3.2% 1 h: mean = 0.4047, SEM = 0.0289, *n* = 9 and Sev 3.2% 2 h: mean = 0.4633, SEM = 0.0233, *n* = 9), which was also observed in the controls (testing phase: saline: mean = 0.6771, SEM = 0.0402, *p* = 0.0005, *n* = 13 and familiarization phase: saline: mean = 0.472, SEM = 0.0333, *n* = 13) (Figure 2c). Similarly, the relative time spent exploring the objects was not significantly different between the groups (Figure 2d).

Taken together, these data suggest that neonatal rat exposure to clinically relevant concentrations of sevoflurane resulted in permanent, long-term detrimental effects on recognition memory.

### 2.4. Sevoflurane Exposure Compromised Hippocampal-Dependent Memory

To specifically test for hippocampal-dependent memory, we subjected the animals to contextual fear-conditioning. Specifically, the animals were placed in a chamber with metal bars at the bottom, contained in an isolated apparatus, and left to freely interact for 2 min (habituation phase), where the freezing time and rearing were monitored. Following the habituation phase, they received three foot-shocks, each one separated from the other by a 2 min interval. The next day, the animals were placed in the same chamber for 7 min, where they did not receive any shock, but their freezing and rearing times were again monitored and compared to the habituation phase (Figure 2e). Interestingly, both animal groups exposed to sevoflurane did not exhibit significant freezing during the day 2 phase (Sev 3.2% 1 h: mean = 0.1613, SEM = 0.0225, *p* = 0.3145, *n* = 9 and Sev 3.2% 2 h: mean = 0.1968, SEM = 0.0199, *p* = 0.1076, *n* = 7) compared to the habituation phase (Sev 3.2% 1 h: mean = 0.06997, SEM = 0.0408, *n* = 9 and Sev 3.2% 2 h: mean = 0.0654, SEM = 0.0217, *n* = 7), in contrast to their control counterparts (testing phase: saline: mean = 0.3383, SEM = 0.0637, *p* = 0.0001, *n* = 9 and habituation phase: saline: mean = 0.06321, SEM = 0.0099, *n* = 9). Specifically, the control group exhibited significantly more freezing on the second day in anticipation of the shock based on their previous day’s experience (Figure 2f).

With regard to the rearing (both front feet above the ground exhibiting an extended posture—representing an exploratory behaviour), we did not find any statistically significant differences between the ratio in the animals exposed to sevoflurane during the testing phase (Sev 3.2% 1 h: mean = 0.1565, SEM = 0.0363, *p* = 0.9451, *n* = 9 and Sev 3.2% 2 h: mean = 0.1102, SEM = 0.0125, *p* = 0.8028, *n* = 9) compared to the habituation phase (Sev 3.2% 1 h: mean = 0.1907, SEM = 0.0257, *n* = 9 and Sev 3.2% 2 h: mean = 0.1583, SEM = 0.0272, *n* = 9). These were statistically insignificant from their control counterparts (testing phase: saline: mean = 0.3383, SEM = 0.0637, *p* = 0.1577, *n* = 9 and habituation phase: saline: mean = 0.2121, SEM = 0.0189, *n* = 9) (Figure 2g). Even though there was no statistically significant difference in this parameter, we noted that the controls showed a more pronounced decreasing trend toward the testing phase in comparison to the habituation phase.

These data suggest that the animals exposed to sevoflurane exhibited a compromised hippocampal-dependent memory, as shown by their decreased freezing ratio compared to the control animals.

### 2.5. DEX IP Pre-Treatment Resulted in a Significant Respiratory Depression

We next asked if a pre-treatment with DEX, administered intraperitoneally (IP), may prevent the detrimental effects on learning and memory seen in sevoflurane-exposed animals. Specifically, we had previously demonstrated that a subcutaneous application of DEX could significantly rescue memory deficits [27], but here we employed an IP mode of administration. The animals were injected with saline (controls) or DEX IP and continuously monitored for an hour. Respiratory rate was monitored every 10 min for the first 30 min after the injection. We observed that the breathing rate in animals injected with DEX IP decreased significantly (t = 0: mean = 23.25, SEM = 0.8539, *p* < 0.0001, *n* = 40) compared to that in their control counterparts (t = 0: mean = 33.51, SEM = 0.6425, *n* = 39) (Figure 3a).

We also monitored oxygen saturation (SpO_2_) every 30 min for 1 h following the injections. For this parameter, there were no significant differences between the groups (Figure 3b); however, for the DEX groups, and unlike their saline-injected controls, a pattern was observed where their SpO_2_ spiked at t = 30 and decreased by t = 60 (Figure 3b).

Overall, these data suggest that the IP mode of DEX administration exerts an important impact, causing respiratory depression that might possibly be accountable to some extent for compromising neuronal health.

### 2.6. DEX Pre-Treatment Via IP Mode of Administration Impacted the Exploratory Behaviour

To assess the levels of “anxiety”, locomotor skills, and “willingness” to explore a new environment, the animals underwent open field testing as shown above. The parameters analyzed were mobile time (Figure 4a), immobile time (Figure 4b), immobile ratio (Figure 4e) and total distance travelled within the box (Figure 4d), where averages per area were compared between the groups using one-way ANOVA statistical testing. We found no significant differences in any of these four parameters within the different groups. The other two parameters analyzed were time spent in each area (Figure 4c) and the total number of entries made to each area (Figure 4e). In the sevoflurane-only exposed groups, the preference for each area was corners > walls > inner areas > centre, indicative of a constant pattern of anxiety-like behaviour where the animals preferentially stayed in the most isolated areas. Interestingly, the animals pre-treated with DEX and later exposed to 2 h of sevoflurane did not show any increased time spent in the corners, in complete contrast to their control sevoflurane-only counterparts (Figure 4c).

When examining the heatmaps, we observed distinct differences in patterns between the groups. Specifically, the animals pre-treated with DEX and subsequently exposed to sevoflurane for 1 h (Sev + dex) equally explored all of the corners (Figure 4i), while the sevoflurane-only rats explored only two corners preferentially (Figure 4h). Similarly, the animals pre-treated with DEX and subsequently exposed to sevoflurane for 2 h explored three of the corners (Figure 4l), in contrast to their sevoflurane-only counterparts that preferentially stayed in one corner. Interestingly, when comparing control (saline) animals to DEX-only animals, we saw that although the DEX animals spent more time in two corners (Figure 4i) and the controls explored three corners (Figure 4g), the preferred corners in the DEX-only groups were opposed diagonally. Thus, the DEX animals had to travel longer distances and explore more in order to get to the two preferred corners.

In summary, the IP pre-treatment with DEX did not appear to significantly modify locomotor behaviour, but it decreased the “reluctance” and “hesitation”, and thus, these animals exhibited a more explorative behaviour.

### 2.7. DEX Pre-Treatment Did Not Significantly Impact Spatial Memory

To further assess if IP-DEX pre-treatment affected spatial memory, we used MWM for the DEX pre-treated animals and those that were exposed to sevoflurane only. Average latency (Figure 4m), distance swum (Figure 4n), and swimming trajectories for all quadrants (Figure 4q–t) were measured for each trial. For allthe conditions, we saw a significant decrease in latency, suggesting that the pre-treatment had neither improved nor worsened spatial memory. We did, however, note that in sevoflurane-only treated animals, the distance travelled was not decreased significantly over time, in contrast to the control (saline) animals.

Interestingly, when we examined the swimming patterns, we noted that the trajectories of the DEX pre-treated animals only seemed more straightforward for quadrants one (Figure 4q) and two (Figure 4r), not being so evident for quadrants three (Figure 4s) and four (Figure 4t).

We next sought to assess if the consolidation of the spatial learning occurred and could be maintained over time in DEX pre-treated animals, as it did for the sevoflurane-only exposed animals. Indeed, we observed a similar pattern where both latency (Figure 4o) and distance swum (Figure 4p) did not significantly differ for days 14, 28, and 56, suggesting that the memory was successfully recalled for all groups.

These data suggest that DEX pre-treatment did not significantly impact spatial memory, but it could not fully rescue the “hesitant” and “unsure” phenotypes seen in sevoflurane-treated animals.

### 2.8. DEX Pre-Treatment Did Not Prevent Recognition Memory Deficits Caused by Sevoflurane Exposure

As we have shown above, neonatal exposure to sevoflurane caused detrimental effects on recognition memory. Therefore, we next assessed if DEX pre-treatment could prevent those effects. We saw that the pre-treatment did not significantly improve recognition memory for either of the sevoflurane exposure times and that, in fact, the animals exposed to DEX alone showed some cognitive dysfunction as well (testing phase: DEX IP: mean = 0.6035, SEM = 0.0576, *p* = 0.9481, *n* = 10 and habituation phase: DEX IP: mean = 0.5126, SEM = 0.01395, *n* = 10) (Figure 5a). When looking at the object that changed location in the box, the DEX-only exposed animals were similar to their control counterparts, being able to recognize the change in location. However, in the animals exposed to sevoflurane, regardless of DEX pre-treatment, the recognition of this change in location did not statistically differ from the familiarization phase, although all groups showed a trend toward spending more time with the object in the new location (Figure 5c). The relative time interacting with the object did not statistically change between the groups for object recognition (Figure 5b) or change in location (Figure 5d).

These data suggest that IP DEX-pre-treatment not only failed to rescue the recognition deficits seen in animals exposed to sevoflurane, but it also caused a recognition deficit of its own when administered alone, suggesting that DEX may not be neuroprotective for this type of memory.

### 2.9. DEX IP Pre-Treatment Did Not Prevent Hippocampal-Dependent Memory Deficits

To further assess the neuroprotective properties of DEX against sevoflurane-induced toxicity in learning and memory, we pre-treated our animals with IP-DEX and assessed hippocampal-dependent memory using contextual fear-conditioning. Interestingly, the animals exposed to DEX only resembled the controls, as they exhibited significant freezing time during the testing phase (DEX IP: mean = 0.303, SEM = 0.0376, *p* = 0.0347, *n* = 10) when compared with the habituation phase (DEX IP: mean = 0.1249, SEM = 0.0366, *n* = 10) (Figure 5e).

On the other hand, the animals pre-treated with DEX and later exposed to sevoflurane did not significantly anticipate the shock. These data suggest that their hippocampal-dependent memory had been compromised.

When looking at the rearing ratio, unlike the trend seen in the controls, the rats exposed to DEX only did not appear to have a reduced number of rearings in the testing phase (Figure 5f). This might suggest a more explorative behaviour indicative of a less fearful or anxious phenotype [56], despite the increased freezing time.

### 2.10. Sevoflurane and Dexmedetomidine Differentially Affected Patterns of Gene Expression

The above data demonstrate that sevoflurane affected learning and memory and that the observed changes lasted for as long as the animals’ behaviour was assessed. To determine if these long-term changes may either directly or indirectly involve gene expression, we adopted a Nanostring approach. Specifically, the animals were exposed to sevoflurane 3.2% for 2 h with or without IP-DEX pre-treatment or just DEX. Twenty-four hours after the exposure, the animals were sacrificed, and the hippocampus was collected for RNA isolation.

Our Nanostring panel could analyze 770 genes, where seven housekeeping genes (Aars, Cnot10, Csnk2a2, Fam104a, Lars, Tbp, Xpnpep1) were used to normalize the gene expression patterns. For the sake of simplicity, we show here only those genes where expression patterns were significantly modified following the anesthetic exposure. In Figure 6, we categorize these genes according to their functional significance: neurotransmission (Figure 6a), glial homeostasis (Figure 6b), pathways regulating glial function (Figure 6c), and inflammation and cell stress genes (Figure 6d).

Similarly, we created a heatmap with all of those genes and created a tree that compared the most similar treatment conditions to each other in terms of gene expression patterns (Figure 6e). Interestingly, animals exposed to sevoflurane that were pre-treated with DEX seemed to have a gene profile more similar to that of the control samples, followed by DEX-only exposed animals. The animals exposed only to sevoflurane differed the most from the control animals.

For further analysis of each of the conditions, when individually plotted against each other, only those genes that exhibited changes greater than 1.5 fold in any of the conditions were kept in the plots (Figure 7). The graphs show the expression patterns compared between the groups as follows: control versus DEX (Figure 7a,b), control versus sevoflurane 3.2% 2 h (Figure 7c,d), control versus DEX IP pre-treated sevoflurane-exposed animals (Figure 7e,f), DEX versus sevoflurane (Figure 7g,h), sevoflurane versus Sev + dex (Figure 7i,j) and DEX versus Sev + dex (Figure 7k,l).

We selected and depicted genes of hypothesized potential relevance in the cellular context that were upregulated (Figure 8) or downregulated (Figure 9) by sevoflurane so that the functionality of those genes could be deduced from their ascribed functions.

In the context of genes that were upregulated following sevoflurane exposure, we discovered modifications both at the pre- and postsynaptic levels. One of the most dramatic changes that occurred at the presynaptic level was in Slc17a7, which encodes for the glutamate transporter required for excitatory synaptic transmission [57,58]. Similarly, at the postsynaptic level, we found significant perturbation of the Tbr1 gene, which is a transcription factor involved in promoting axonal migration, long-term potentiation, synaptic plasticity, and long-term memory [59]. Genes such as Dlx 1 and 2, Mapk8, Camk2a, Ptk2b, and Rasgrp1 were all upregulated; these have generally been ascribed postsynaptic functions and are involved in the upstream signalling pathway underlying synaptic plasticity.

Furthermore, other transcriptional factors that were also upregulated are responsible for cell survival (Npas2), neurogenesis (Eomes), repolarization of the cell during an electric stimulus (Kcnv1), and long-term neuronal plasticity and memory (Egr1). At the presynaptic level, Egr1 is also known to modulate the expression of Vamp2, which mediates synaptic exocytosis. Additionally, the gene that codes for the transport of pregnanolone sulphate (PregS) through the blood–brain barrier and is also implicated in synaptic plasticity was found to be upregulated in sevoflurane-exposed animals. Interestingly, the expression patterns of all of these genes were brought back closer to the control levels by DEX, although animal pre-treatment with this anesthetic did not completely reverse the gene expression changes induced by sevoflurane.

On the other hand, when we examined genes that were downregulated following sevoflurane exposure, we found a considerable number of them, including Plp1, Mag, Mbp, Mog, Ugt8a, and Slc17a6. An altered expression of these genes has previously been implicated in myelination, cell death, and inhibition of axonal outgrowth and neuronal plasticity. Other genes that were downregulated include Aldh1a, known to be involved in detoxification, and Pecam1, involved in the promotion of vascular repair mechanisms and blood–brain barrier integrity after multiple sclerosis (MS)-like insults and either directly or indirectly involved in myelination. Even though DEX pre-treatment resulted in the upregulation of genes that promote myelination, those that promote synaptogenesis, neuroplasticity, and inhibition of apoptotic mechanisms, such as Adamts1, Nefh or Lgals3, were similarly affected by both DEX and sevoflurane exposure.

## 3. Discussion

Anesthetic-induced neurotoxicity has long been debated for a variety of agents, and their effects on long-term learning and memory are well-documented in animal studies. In parallel, the search has continued for those adjuvants that could potentially mitigate the anesthetic-induced effects on cellular viability, growth, synaptic connectivity, learning, and memory. DEX has since been appreciated for its ability to reduce the incidence of emergence agitation [60,61,62] and emergence delirium [63,64,65] and to decrease sevoflurane requirements while maintaining the state of anesthesia [66,67]. In this study, we demonstrated its ability to improve “hesitant and anxious behaviours”; however, its intraperitoneal administration was not sufficient to improve cognitive deficits induced by sevoflurane exposure. Here. for the first time, an in-depth analysis was performed to study the effects of both sevoflurane and dexmedetomidine exposure on short-term gene expression that may correlate with long-term cognitive changes and possible neuroprotective mechanisms.

Previous studies have shown the long-term effects of sevoflurane on learning and memory. For instance, sevoflurane exposure during the early stages of life has been reported to cause learning and memory deficits [21,23,24,29,30] in rodents. However, in some murine [68,69,70] as well as human studies [16,17,18,19], no effect on cognition was reported, while in other cases, an improvement in certain aspects of cognition was observed in rodents [71,72]. Interestingly, in studies reporting a long-term cognitive decline, sevoflurane was infused with oxygen concentrations ranging from 30–60% [27], while those studies demonstrated no detrimental effects on the memory used sevoflurane mixed with pure oxygen, as is sometimes the case in clinical practice [73,74,75]. Although there is a disagreement as to how much oxygen should be provided during anesthesia, some studies claim that excessive oxygen administration may cause atelectasis (lung collapse caused by excessive oxygenation) [76] or increased oxidative stress [77]. Others studies, however, have claimed that, hyperoxia could in fact be beneficial to improve patients’ outcomes after anesthesia [78,79,80], as such treatment has also been shown to prevent apoptosis while promoting neurogenesis [81,82]. In our study, we used 75% oxygen with a clinically relevant sevoflurane concentration for neonates [83] and exposure times that are typical for most medical or surgical procedures required for newborns [84].

Here, we showed that different behavioural aspects were impacted by neonatal sevoflurane exposure. Our data are thus consistent with what has been previously reported in the literature for deficits in object recognition [23,29,85] and hippocampal-dependent memory [86,87,88] caused by sevoflurane exposure. However, in this study, we did not observe significant differences in spatial memory as previously reported in the literature [21,22,24]. This discrepancy may be due to the fact that in other studies, the neonates were exposed repeatedly to sevoflurane, which may be less clinically relevant than the single exposure used here. Nevertheless, here, we observed several other erratic behaviours that had not been reported previously, concomitant with aberrant swimming patterns where the animals often missed the targets in their first attempts. It is interesting to note that by looking solely at the time to reach the platform (with animals swimming at a faster pace), this observation would have failed to take into account the distance that the animals travelled to reach their targets.

Furthermore, in this study, we looked not only at spatial memory but also at the consolidation phase of the aberrant behavioural repertoire. Ultimately, we noted a more hesitant and reluctant nature of behaviour in animals exposed to sevoflurane for 2 h, which we would anticipate being further enhanced with longer exposure time periods or repeated anesthetic treatments.

Dexmedetomidine has gained some notoriety in the past years as a possible neuroprotectant agent. Studies have looked at putative mechanisms underlying its neuroprotective effects at the molecular level and shown that dexmedetomidine may downregulate proteins implicated in apoptosis [13,25,51,89], promote neuronal outgrowth [12,55], modify mitochondrial morphology and function [52,55] and decrease inflammatory cytokines [53]. Behavioural studies where DEX was used as a pre-treatment prior to sevoflurane exposure have found improvements in swimming latency [24,90] and object recognition [23,27]. However, in our study, we did not see an improvement in any of these parameters; rather, we found DEX to be detrimental with respect to recognition memory. The reason why this may not have been reported in previous studies could be due to the fact that the test most often used to assess memory improvements is the Morris water maze test, which did not significantly differ betwen the controls and treated groups in our study, consistent with some previous reports. Secondly, the concentration of DEX used in previous studies was lower than the one attempted here for IP administration. It is therefore possible that concentrations above 20 µg/kg of IP DEX may be detrimental toward certain types of memories and thus may not be neuroprotective.

Previous studies aimed at identifying genes affected by anesthetics have focused primarily on factors underlying apoptosis, inflammation, or mitochondrial function instead of performing a more in-depth screening. Here, we performed Nanostring analysis in order to better analyze those genes that may account for anesthetic-induced cytotoxic effects underlying learning and memory. Specifically, we analyzed 770 genes related to neurotransmission and glial homeostasis, regulation, and activation, as well as inflammation and cell stress, giving us a better perspective on potential genes that might have been impacted after neonatal exposure to anesthetics—especially those that could account for learning and memory deficits. We further focused on those genes that had previously been implicated in synaptic plasticity, learning, and memory, and also the ones involved in neuronal and axonal degeneration. Specifically, we observed differences in 17 genes that had previously been implicated in synaptic plasticity and memory formation and in 22 genes that are associated with demyelination, neuroplasticity, and cellular toxicity.

In this study, we saw a pronounced upregulation in the expression of the vesicular glutamate transporter 1 (V-Glut1/Slc17a7) after sevoflurane exposure. The overexpression of this gene in turn may lead to an excessive glutamate release into the synaptic cleft, thus triggering glutamate-induced excitotoxicity [91]. Glutamate-induced excitotoxicity promotes increased cell death caused by an excessive release of this neurotransmitter by either neuron or glial cells [92]. The glutamate-induced excitotoxicity has been reported after traumatic brain injury [93], ischemia [92], and different neurodegenerative pathologies such as Alzheimer’s [94], Parkinson’s [95], and Huntington’s [96] disease. This excessive glutamate may activate the NMDAR in the postsynaptic membrane, leading to a prolonged influx of Ca^2+^ into the postsynaptic neuron. Excessive un-sequestered Ca^2+^, by intracellular reuptake organelles, may in turn activate Ca^2+^-activated proteases, which may sculpt either dendritic or axonal branching, thus leading to a loss of synaptic structures [97]. Such a mechanism is known to function during Wallerian degeneration following axonal and nerve injuries [98]. Moreover, higher levels of Ca^2+^ may, in turn, result in an augmented calcium signalling, eventually leading to a gradual loss of synaptic function and ultimately to cell death [99,100]. A loss of dendritic branches, axons, synaptic structures, neurons, or glia in turn may result in the progressive memory deterioration seen in different neurodegenerative diseases [101]. An upregulation of V-Glut1 may thus underlie changes to all of the above cellular structures and be responsible for the perturbation of recognition and contextual memories after sevoflurane exposure. Taking into consideration the fact that DEX pre-treatment did not completely prevent sevoflurane-induced upregulation of this gene, our data suggest that V-Glut1 could potentially be one of the targets for both agents tested in this study.

Interestingly, another gene whose expression levels were significantly increased after sevoflurane exposure turned out to be the T-box brain gene 1 (Tbr1), which has been implicated in: (a) axonal migration via the regulation of reeling (Reln) [102] and (b) controlling N-methyl-D-aspartate (NMDA) receptor 2b (Grin2b) expression, which is essential for learning and memory [59]. Tbr1 is regulated by Ca^2+^/calmodulin-dependent protein kinase II (Camk2a), believed to serve as a trigger for memory formation [103]. Similarly, Camk2a is regulated by calcium binding, which is controlled by a cascade of proteins, such as RAS guanyl nucleotide-releasing protein 1 (Rasgrp1), Ras, and Mitogen-activated protein kinase 8 (Mapk8); these were also upregulated by sevoflurane. Interestingly, after DEX pre-treatment, Trb1 expression was brought back to the control levels. However, exposure to DEX alone resulted in an increased expression of Tbr1 on its own, although this effect was less pronounced than for sevoflurane. Regarding the other genes involved upstream, only Camk2a was slightly modified by DEX exposure, which could explain the lack of complete protection against the deterioration of the long-term memory of adult animals. Similarly, other genes implicated in learning and memory, such as the early growth response protein 1 (Erg1) [104], distal-less homeobox 1 and 2 (Dlx1 and 2) [105], and neurogenesis Tbr2 (Eomes) [106,107], were also upregulated by sevoflurane exposure and partially rescued by DEX pre-treatment. The effects on Tbr1 expression have been previously reported in the literature with concentrations of 10 µg/kg of IP DEX [108]. It therefore stands to reason that DEX had the potential to bring back the expression level of these genes to their baseline, and this may thus underlie its neuroprotective effect; this would need to be tested further experimentally. Other genes whose expression was affected by sevoflurane exposure are the blood–brain barrier transporter of PregS that promotes NMDAR activity [109], further increasing the influx of calcium into the postsynaptic neuron, and the intracellular tyrosine kinase Pyk2 (Ptk2b) that has been reported to be involved in the modulation of excitatory synapses in the hippocampus [110]. We found that both genes were upregulated by sevoflurane exposure and only partially rescued by DEX pre-treatment, further underscoring the ability of DEX to be potentially neuroprotective.

Some additional indicators of DEX’s neuroprotective potential are also evident in this study. For example, a disintegrin and metalloproteinase with thrombospondin motifs (Adamts1) is known to promote synaptogenesis and neuroplasticity [111,112] and inhibit mitochondrial apoptosis [113]; its expression was upregulated by DEX pre-treatment prior to sevoflurane exposure. Similarly, the stem cell factor Kitl has been implicated in promoting synaptogenesis and neuroplasticity through P13K activation [114], which we found to be significantly increased by DEX alone, but not following sevoflurane exposure. Furthermore, galectin 3 (Lgals3) has been shown to promote neurogenesis after brain injury through IGF1 [115], and these expression levels were enhanced following DEX pre-treatment. Additionally, the Aldehyde Dehydrogenase 1 Family Member A1 (Aldh1a1) has been shown to promote detoxification and diminish ROS production [116]; its expression was significantly decreased by both DEX and SEV when applied individually, but significantly upregulated following the DEX pre-treatment. The fact that the exposure to both anesthetics resulted in a more pronounced upregulation of this gene could suggest an additive effect that may contribute toward DEX’s role in rescuing cellular toxicity when both anesthetics are applied conjointly.

Finally, we report here that several genes, previously implicated in myelination, were also altered following sevoflurane exposure. For example, sevoflurane decreased the expression of: (a) Proteolipid Protein 1 (Plp1), which plays a role in oligodendrocyte differentiation [117], (b) the myelin-associated glycoprotein (Mag), an activator of RhoA that promotes cell death, myelination and axon degradation [118], (c) the Myelin basic protein (Mbp), which increases the permeability of the blood–brain barrier [119], (d) the Myelin oligodendrocyte glycoprotein (Mog), whose expression is decreased in Huntington’s disease [120], and (e) the UDP Glycosyltransferase 8 (Ugt8a) involved in changing the composition of the sphingolipids of the membrane [121]. Therefore, sevoflurane exposure could correlate with an altered myelination profile, even though this process is yet to occur in a neonate. Myelination is necessary for all mammalian neuronal networks to fully function, and some diseases, such as MS, promote aggressive demyelination that results in the disruption of nerve fibres, leading to symptoms such as sensory disorders, visual damage, or cognitive and emotional impairment [122]. The fact that some genes implicated in myelination were found to be downregulated after sevoflurane exposure is interesting, taking into consideration that myelination does not begin to occur in the rats until P10 [123], when these pups were exposed at P7, and their gene expression quantified at P8.

Overall, our study examined a host of genes that may be altered following anesthetic exposure. Of particular interest are the genes involved in proper brain development, synaptic transmission, plasticity, and learning and memory. Further exploration of these genes and their direct impact on neuronal transmission and plasticity will help elucidate how anesthetics may induce their long-term effects on the nervous system.

## 4. Materials and Methods

### 4.1. Experimental Animals

Pregnant rats (Sprague–Dawley strain code 400) were purchased from a commercial breeder (Charles River Laboratories, Senneville, QC, Canada). These rats were examined throughout their pregnancy from E16.5 until pups’ birth (post-natal day zero—P0). Rats were kept in a conventional room set at 25 °C on a 12 h light/dark cycle from 7 am to 7 pm, and rats were fed ad libitum.

### 4.2. Dexmedetomidine Treatment

The pups were injected at P7 with 25 µg/kg of dexmedetomidine intraperitoneally (IP) using a 0.1 mL volume, whereas the control animals received a corresponding volume of saline (sodium chloride 0.9% B Braun, Mississauga, ON, Canada). The animal’s body temperature was maintained by a heated blanket and nitrile gloves filled with warm water, which was replaced every 20 min. The respiratory status of the animal was monitored by counting the number of breaths per minute every 10 min after the first injection for 30 min, to ensure the health of the pup. Similarly, oxygen saturation (SpO_2_) was monitored (Nonin Medical, Minneapolis, MN, USA) every 30 min after the first injection for a period of 1 h to further ensure the pup’s health.

### 4.3. Sevoflurane Exposure

After a 1 h pre-treatment either with saline (control) or DEX, the pups were placed in an anesthetic chamber for sevoflurane exposure, as a sevoflurane-medical air-oxygen gas mixture (75% O_2_), vaporized using a Datex-Ohmeda Aestiva/5 vaporizer. The sevoflurane concentrations were monitored with a GE Healthcare Gas Analyzer. The sevoflurane concentration used was 3.2% for 1 or 2 h at P7 [27]. The pups used for molecular testing were euthanized 24 h after anesthetic exposure using a decapitation method. Brains were isolated from the pups, and the hippocampus was carefully separated and stored at −80 °C.

The pups randomly selected for behavioural testing were ear-notched at P14 and weaned in cages of 2 to 3 animals, separated by their sexes. Animals were kept undisturbed until P60 and then subjected to various behavioural tests.

**Note:** In order to assess the animals’ behaviours in a more accurate and in-depth way, we used a host of qualitative parameters to explain differences in an animal’s behaviour that are not commonly described in most studies. We incorporated terms such as “erratic” [124,125], “hesitant” [126,127], “unsure”, “undecisive”, exploratory” [127,128], “anxious” [126,127] and “reluctance” to better describe differences between treated and untreated animals.

### 4.4. Morris Water Maze

Animals were placed in a 1.8 m diameter pool of water with a hidden platform at 1 cm below the water surface (25 °C) placed in a room with visual cues on four different walls as previously described [55]. For this study, latency and distances swum were measured with the automated software ANY-Maze (Stoelting Co, Wood Dale, IL, USA) from day 1 to day 5, and the assay was again repeated at days 14, 28, and 56 to assess retention memory.

### 4.5. Novel Object Recognition

Object selection was carried out as previously described [55] using a habituation period of 2 consecutive days where the animal was placed in an empty box for 15 min. After the habituation, the animals went through (1) a familiarization phase where two identical objects were introduced for 5 min, (2) a retention interval where the animals were removed from the box for 5 min, and, (3) a testing phase where the animals were left to interact with a novel object or a familiar object placed at a different location in the box for 5 min. Discrimination index and relative time interacting with the objects were measured using ANY-Maze (Stoelting Co, Wood Dale, IL, USA).

### 4.6. Open Field

As previously described [27], animals underwent open field testing for 15 min in an empty box. An animal’s level of locomotor activity, anxiety, and willingness to explore a new environment were critically assessed. For analytical purposes, the box was divided into 4 analytic areas: corners, walls, inner area, and center. The time spent in each area, entries made to each area, total distance travelled, track paths, and time the rat was immobile were monitored using ANY-Maze (Stoelting Co, Wood Dale, IL, USA).

### 4.7. Contextual Fear Conditioning

The contextual conditioning was assessed in a fear-shuttle apparatus (Imetronic, Pessac, France). Animals were placed in the chamber and allowed to explore the environment for 2 min (habituation phase). The same day, after the habituation phase was completed, animals underwent the conditioning session, where they received 3 foot shocks (2000 ms, 0.5 mA, inter-trial interval 140 s). Twenty-four hours later, rats underwent the test session, where the animals were placed in the same box for 7 min. Freezing behaviour was recorded using a tight infrared frame. Freezing and rearing (standing on hind legs) ratios were assessed and compared between the habituation and the testing phase to evaluate if contextual learning had occurred.

### 4.8. RNA Isolation

Total mRNA was isolated from P8 rat brains specifically from the hippocampal region using the RNeasy micro kit (QIAGEN, Toronto, ON, Canada) according to the manufacturer’s recommendations. The minimum RNA integrity number (RIN) for the samples was 8.6.

### 4.9. Nanostring RNA Expression Quantification

Total mRNA was hybridized and multiplexed with the nCounter^®^ Mouse Glial Profiling Panel, according to the manufacturer’s instructions. Counts for target genes were normalized to housekeeping genes (Aars, Cnot10, Csnk2a2, Fam104a, Lars, Tbp, Xpnpep1) to account for the total RNA contents. The different experimental groups were compared on a one-to-one basis to detect fold changes in the gene expression patterns. A list of the genes in the glial panel can be found at: “https://www.nanostring.com/products/ncounter-assays-panels/neuroscience/glial-profiling/ (accessed on 11 May 2023)”.

### 4.10. Statistical Analysis

All samples were assigned randomly, and the experiments were performed in a single-blinded fashion. Specifically, the observer was unaware of the experimental conditions. Statistical significance tests were performed with GraphPad Prism 8. One-way and two-way ANOVA were used to compare multiple groups, followed by Dunnett’s or Tukey’s multiple comparisons tests for post hoc comparisons. Differences between the means of the two conditions were tested using the two-sided Student’s *t*-test with Welch’s correction. Differences between various data sets were considered significant if appropriate post hoc statistical tests resulted in *p* ≤ 0.05.

## 5. Conclusions

This study not only revealed several changes that occur in an adult animal’s learning and memory as a result of neonatal exposure to sevoflurane but also identified a cohort of genes that could potentially account for those observed perturbations. A direct correlation between anesthetic-induced gene expression and behavioural deficits in learning and memory, however, awaits further experimentation. This study underscores the importance of careful monitoring of young children who are exposed to anesthetics early in their life for changes in behaviour, learning, and memory and highlights the need for additional studies examining gene expression in humans.

## Figures and Tables

**Figure 1 ijms-24-08696-f001:**
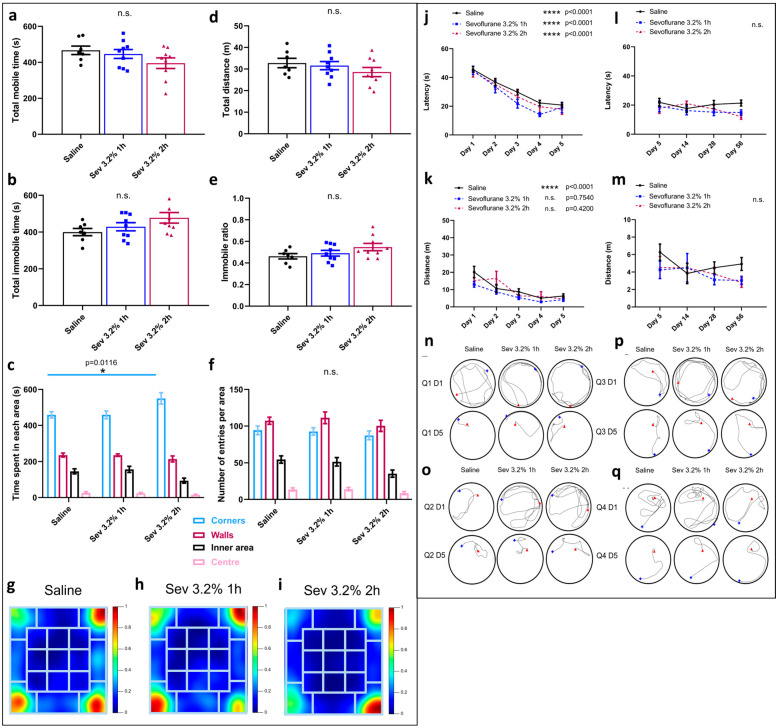
**Sevoflurane exposure resulted in increased hesitant and erratic behaviours. Open field testing:** Representation of (**a**) total mobile time, (**b**) total immobile time, (**c**) time spent in each area, (**d**) total distance travelled, (**e**) immobile ratio, and (**f**) number of entries in each area. * *p* = 0.0116 by two-way ANOVA with Tukey’s post hoc analysis for multiple comparisons. (**g**) Heatmaps representing the average time spent in each area of the box in control animals, (**h**) sevoflurane 3.2% 1 h animals and (**i**) sevoflurane 3.2% 2 h animals (dark blue—less time, dark red—more time). **Morris water maze:** (**j**) average latency and (**k**) average swimming distance between the 4 quadrants needed per day to reach the hidden platform from days 1 to 5, and (**l**) average latency and (**m**) average swimming distance from days 5 to 56. Swimming tracing patterns for days 1 and 5 in quadrants (**n**) 1, (**o**) 2, (**p**) 3, and (**q**) 4 (blue—starting point, red—finish point). **** *p* < 0.0001 by two-way ANOVA with Tukey’s post hoc analysis for multiple comparisons. Bars indicate mean ± SEM. Statistical analysis by two-way ANOVA with Tukey’s post hoc analysis for multiple comparisons **n = 7 for control, *n* = 9 for Sev 3.2% 1 h and Sev 3.2% 2 h**.

**Figure 2 ijms-24-08696-f002:**
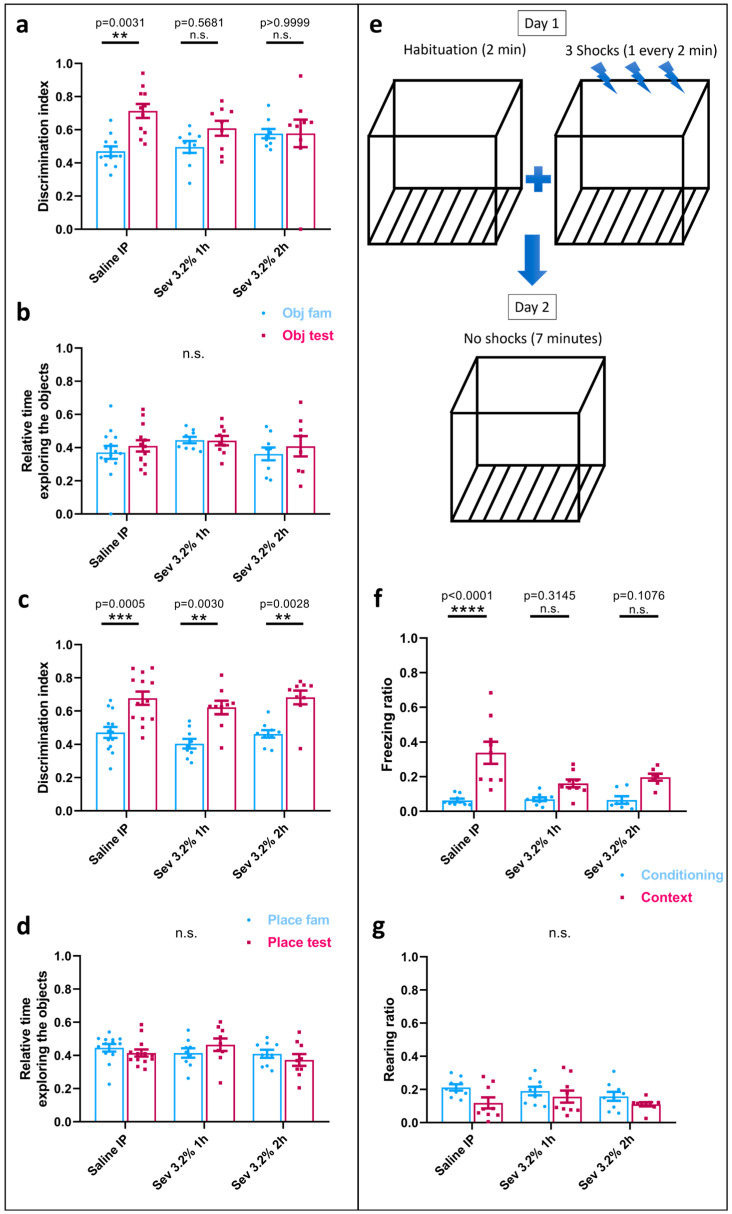
**Sevoflurane exposure compromises recognition memory and hippocampal-dependent contextual memory. NORT:** (**a**) Discrimination index calculated as the time spent with the novel object (TN) divided by the total time spent with both objects (TN + TF). (**b**) Relative time interacting with both objects (TN + TF)/(duration of the phase). (**c**) Discrimination index calculated as the time spent with the object located in a different place (TN) divided by the total time spent with both objects (TN + TF), and (**d**) relative time interacting with both objects (TN + TF)/(duration of the phase). **Contextual fear conditioning testing:** (**e**) Schematic representation of fear conditioning testing, (**f**) freezing ratio, and (**g**) rearing ratio for habituation phase (conditioning) and testing phase (contextual). The ratio is calculated by the time spent either freezing or rearing divided by the total time. Bars indicate mean ± SEM. Statistical analysis by two-way ANOVA with Tukey’s post hoc analysis for multiple comparisons. *n* = 11 for control, *n* = 9 for Sev 3.2% 1 h and Sev 3.2% 2 h for NORT and *n* = 9 for control and for Sev 3.2% 1 h and *n* = 7 for Sev 3.2% 2 h for contextual fear and conditioning test.

**Figure 3 ijms-24-08696-f003:**
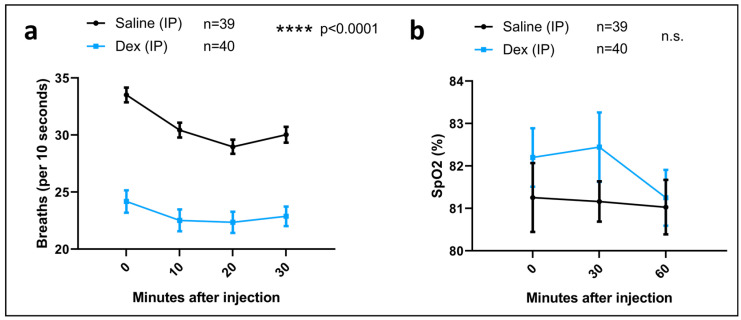
**Intraperitoneal DEX administration diminished respiratory rate but not oxygen saturation.** (**a**) Breaths monitored over time after DEX or saline administration at 0, 10, 20, and 30 min after the injection. Bars indicate ± SEM. *n* = 39–40 animals per condition. (**b**) Oxygen saturation monitored over time after DEX or saline administration at 0, 30, and 60 min after the injection. Bars indicate ± SEM. **** *p* < 0.0001 by two-way ANOVA with Tukey’s post hoc analysis for multiple comparisons. *n* = 39–40 animals per condition.

**Figure 4 ijms-24-08696-f004:**
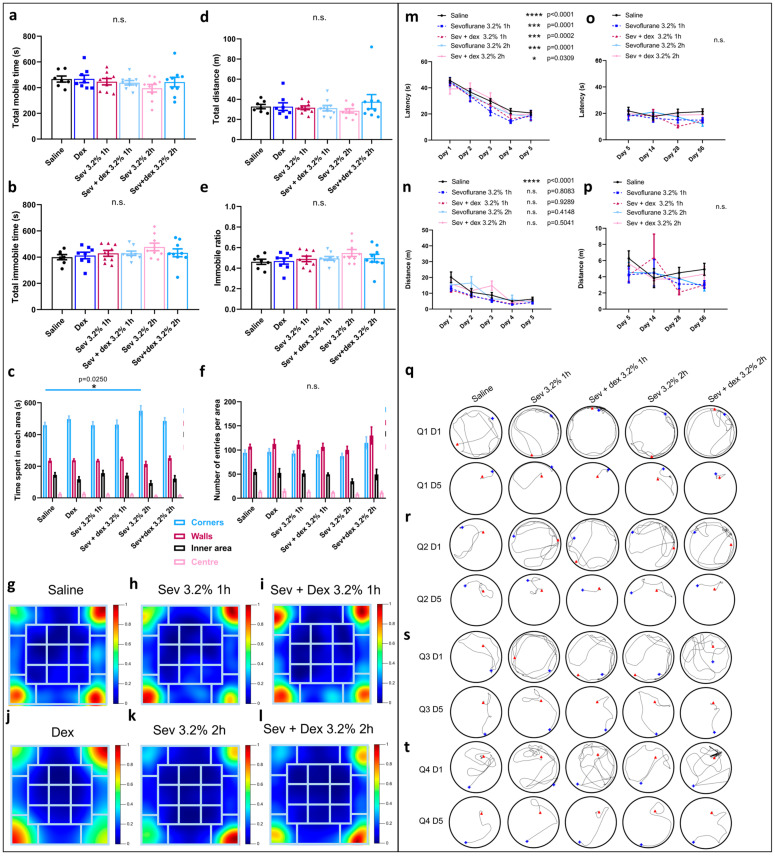
**DEX pre-treatment partially rescues increased hesitant and erratic behaviours caused by sevoflurane exposure**. **Open field testing:** Representation of (**a**) total mobile time, (**b**) total immobile time, (**c**) time spent in each area, (**d**) total distance travelled, (**e**) immobile ratio, and (**f**) number of entries in each area. * *p* = 0.0250 by two-way ANOVA with Tukey’s post hoc analysis for multiple comparisons. (**g**) Heatmaps representing the average time spent in each area of the box in control animals, (**h**) sevoflurane 3.2% 1 h animals, (**i**) sevoflurane 3.2% 2 h animals, (**j**) DEX animals, (**k**) Sev + dex 3.2% 1 h animals, and (**l**) Sev + dex 3.2% 2 h animals (dark blue—less time, dark red—more time). **Morris water maze**: (**m**) average latency and (**n**) average swimming distance between the 4 quadrants needed per day to reach the hidden platform from days 1 to 5. (**o**) Average latency and (**p**) average swimming distance from days 5 to 56. Swimming tracing patterns for days 1 and 5 in quadrants (**q**) 1, (**r**) 2, (**s**) 3, and (**t**) 4 (blue—starting point, red—finish point). Bars indicate mean ± SEM. Statistical analysis by two-way ANOVA with Tukey’s post hoc analysis for multiple comparisons. *n* = 7 for control, *n* = 9 for Sev 3.2% 1 h, Sev 3.2% 2 h, Sev + Dex 3.2% 1 h and Sev + Dex 3.2% 2 h.

**Figure 5 ijms-24-08696-f005:**
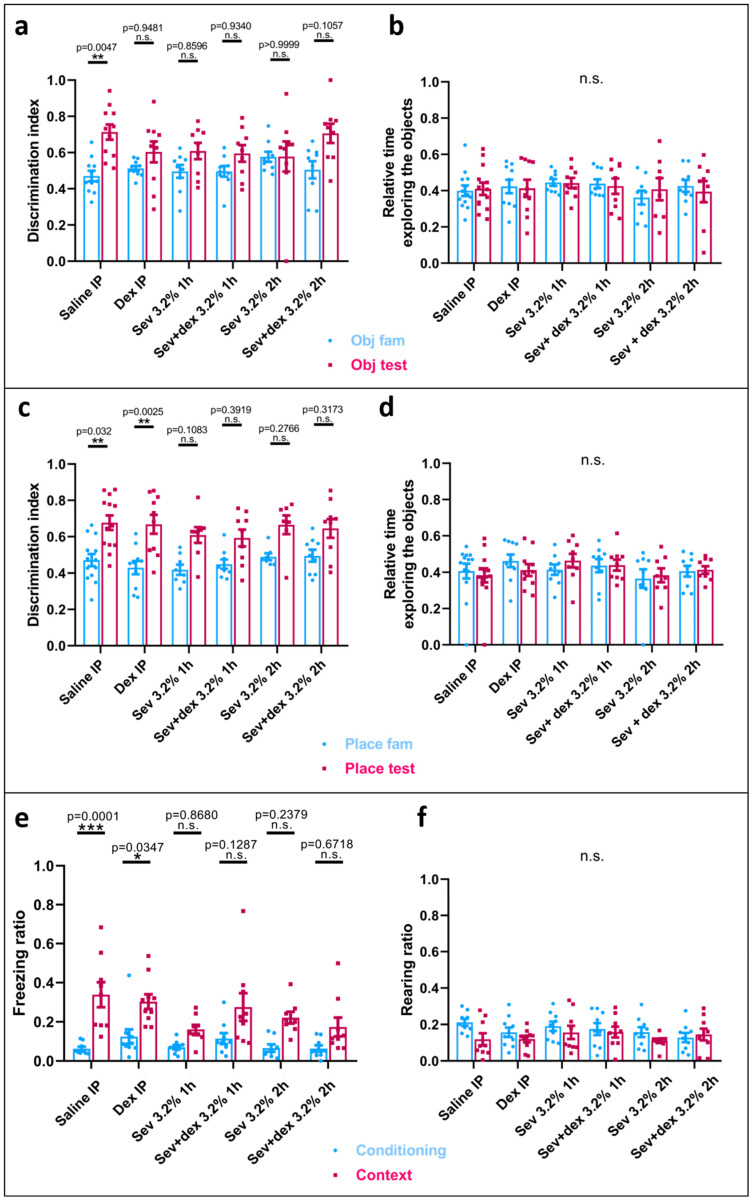
**DEX pre-treatment does not rescue sevoflurane-induced recognition and hippocampal-dependent contextual memory deficits. NORT**: (**a**) Discrimination index calculated as the time spent with the novel object (TN) divided by the total time spent with both objects (TN + TF). (**b**) Relative time interacting with both objects (TN + TF)/(duration of the phase). (**c**) Discrimination index calculated as the time spent with the object located in a different place (TN) divided by the total time spent with both objects (TN + TF) and (**d**) relative time interacting with both objects (TN + TF)/(duration of the phase). **Contextual fear conditioning test**: (**e**) Freezing ratio and (**f**) rearing ratio for habituation phase (conditioning) and testing phase (contextual). The ratio is calculated by the time spent either freezing or rearing divided by the total time. Bars indicate mean ± SEM. Statistical analysis by two-way ANOVA with Tukey’s post hoc analysis for multiple comparisons. *n* = 11 for control, *n* = 9 for Sev 3.2% 1 h, Sev + Dex 3.2% 1 h, Sev 3.2% 2 h and Sev + Dex 3.2% 2 h for NORT, and *n* = 9 for control and for Sev 3.2% 1 h, Sev + Dex 3.2% 1 h and Sev + Dex 3.2% 2 h, and *n* = 7 for Sev 3.2% 2 h for contextual fear and conditioning test.

**Figure 6 ijms-24-08696-f006:**
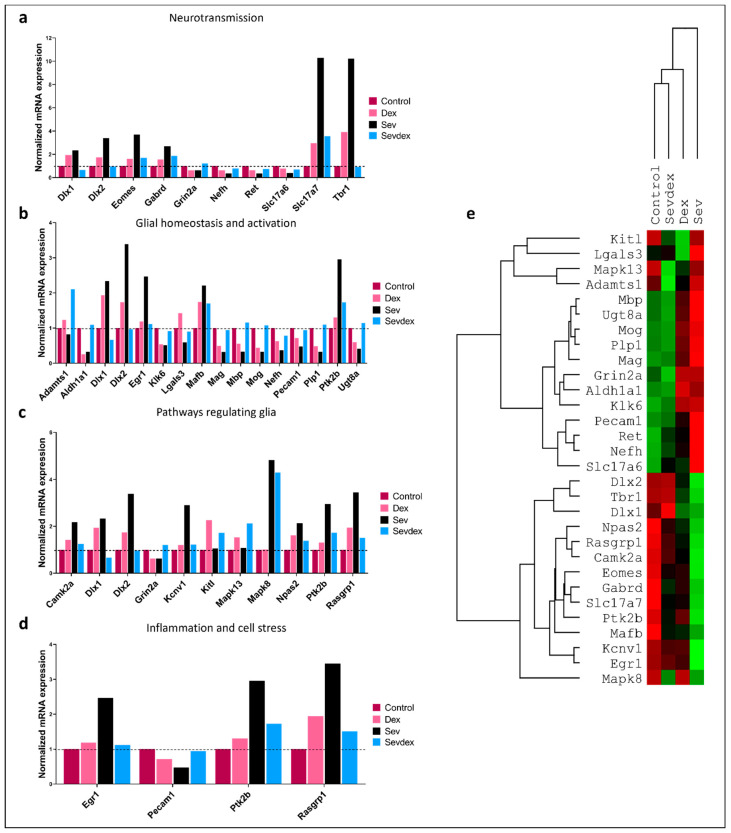
**Sevoflurane and DEX differentially modify gene expression patterns.** (**a**) Analysis of genes involved in neurotransmission, (**b**) glial homeostasis and activation, (**c**) pathways regulating glia and (**d**) inflammation and cell stress, expressed as normalized mRNA expression. (**e**) Heatmap showing representative gene expression patterns between the four different condition groups analyzed. Green indicates upregulation, and red downregulation. *n* = 3 animals per condition.

**Figure 7 ijms-24-08696-f007:**
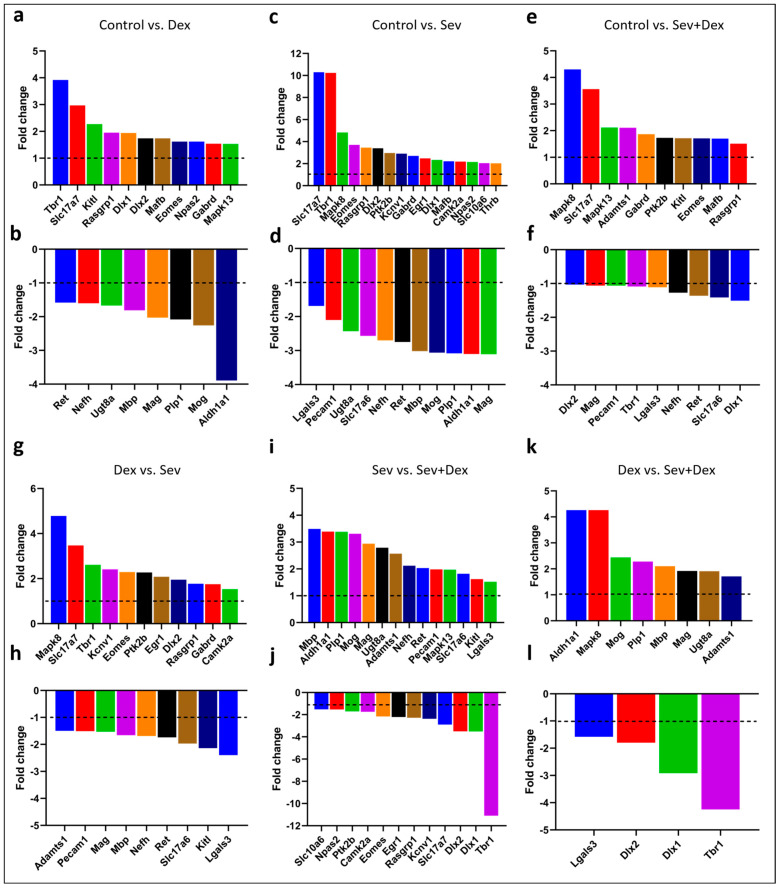
**Representative upregulated and downregulated genes for each condition**. Comparison of gene expression changes (second condition compared to the first) defined as fold-change in (**a**,**b**) control vs. DEX, (**c**,**d**) control vs. sevoflurane, (**e**,**f**) control vs. Sev + Dex, (**g**,**h**) DEX vs. sevoflurane, (**i**,**j**) sevoflurane vs. Sev + Dex, (**k**,**l**) DEX vs. Sev + Dex.

**Figure 8 ijms-24-08696-f008:**
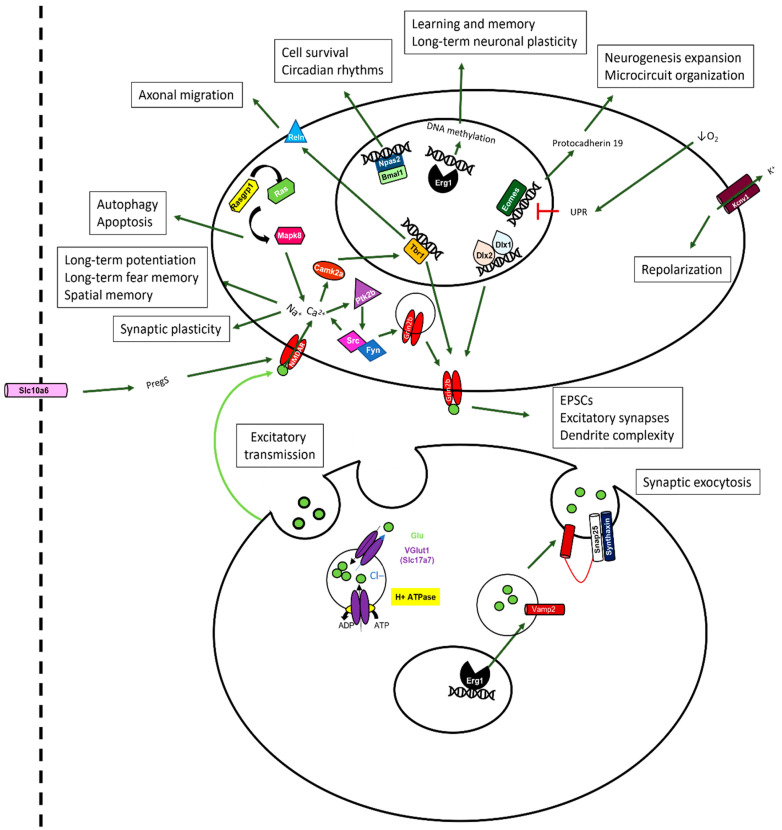
**Schematic representation of the functional implications of the genes upregulated by sevoflurane exposure.** In the presynaptic neuron (bottom), Slc17a7 mediates the influx of glutamate into the vesicles that are implicated in the exocytosis of the neurotransmitter into the synaptic cleft to promote excitatory transmission. This transmitter exocytosis is mediated by Vamp2, which, in turn, is transcriptionally regulated by Erg1. In the postsynaptic neuron (top), the glutamate binding to the NMDAR allows the influx of sodium and calcium into the cell. Calcium can bind to Camk2, which promotes the activation of the transcriptional regulator Tbr1 implicated in controlling the expression of Grin2b. Grin2b regulation is also modulated by the transcriptional factors Dlx1 and 2. The process of exocytosis is also modulated by the Ptk2b/Src/Fyn activation that ultimately phosphorylates Grin2b so it can be inserted into the membrane. Tbr1 is also responsible for the regulation of Reln, involved in axonal migration. The Rasgrp1/Ras/Mapk8 pathway is not only involved in regulating the Camk2a pathway but is also implicated in promoting autophagy and apoptosis. Slc10a6 promotes the entrance of pregnenolone sulphate through the blood–brain barrier that modules the NMDAR activity. Other transcriptional regulators in the nucleus are responsible for promoting cell survival (Npas2), forming the basis for proper establishment of synaptic plasticity and, therefore, learning and memory through DNA methylation (Egr1) and microcircuit organization and neurogenesis expansion (Eomes). As the postsynaptic neuron depolarizes due to sodium entry through the NMDAR, Kcnv1 causes the efflux of K+ out of the cell to promote repolarization. EPSCs = Excitatory postsynaptic potentials.

**Figure 9 ijms-24-08696-f009:**
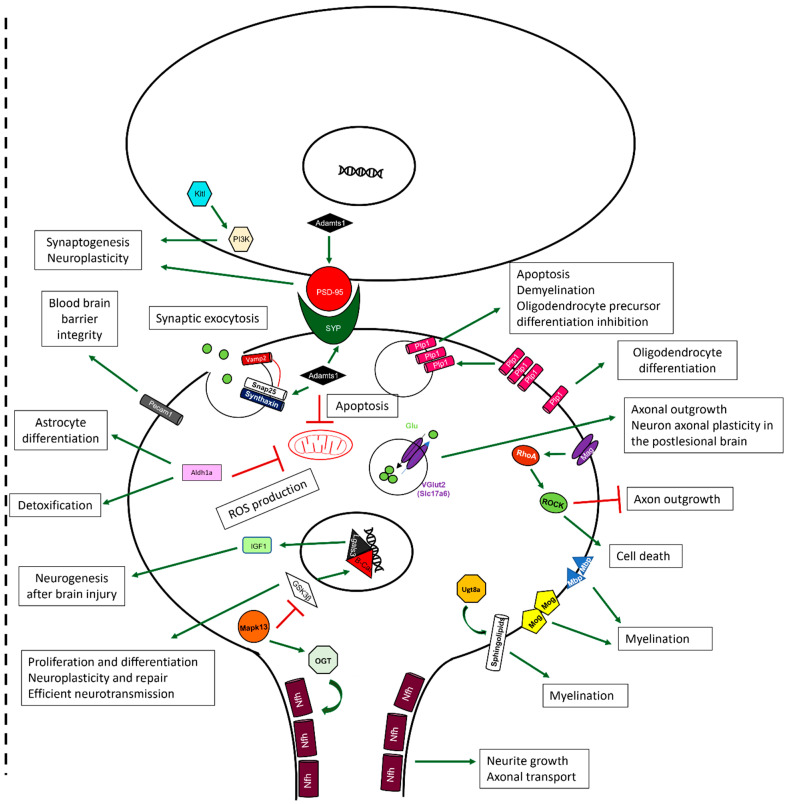
**Schematic representation of the functional implications of the genes downregulated by sevoflurane exposure.** In the presynaptic neuron (bottom), there are several genes involved in the control of normal myelination processes. Plp1 is necessary for oligodendrocyte differentiation, which is required during the myelination process. Ugt8a is responsible for the correct synthesis of the sphingolipids that form part of the cellular membrane. Mog, Mbp, and Mag are proteins responsible for the correct myelination of the axons. The latter is also implicated in regulating the RhoA/ROCK pathway, which is involved in promoting cell death and inhibiting axon outgrowth. Other proteins that are involved in axonal outgrowth are Vglut2, which also plays a role in promoting neuronal and axonal plasticity in the postlesional brain, and Nfh, also implicated in axonal transport. Nfh is regulated by the Mapk13/OGT pathway. Mapk13 also inhibits the function of GSK3β, which is responsible for: (a) activating the Lgals3/β-Cat transcriptional complex involved in the transcription of IGF1, which promotes the neurogenesis after brain injury, (b) proliferation and differentiation, and (c) efficient neurotransmission, neuroplasticity, and repair. Aldh1a is known to diminish ROS production and plays a role in detoxification and astrocyte differentiation. Adamts1 is thought to inhibit apoptosis, promoting vesicular exocytosis and synaptogenesis. Pcam1 is responsible for maintaining blood–brain barrier integrity. In the post-synaptic neuron (top), the Kitl/PI3K pathway mediates synaptogenesis and neuroplasticity.

## Data Availability

The data presented in this study are available on request from the corresponding author.

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
