# Peer review of "Sevoflurane Exposure in Neonates Perturbs the Expression Patterns of Specific Genes That May Underly the Observed Learning and Memory Deficits"

_ijms, 2023, doi:10.3390/ijms24108696_

Round 1

Reviewer 1 Report

Reviewer comments & suggestions

Nerea Jimenez-Tellez and the group have discussed the “Sevoflurane exposure of the neonates perturbs the expression patterns of specific genes, which may underly the observed learning and memory deficits” in this present research article. Exposure to commonly used anesthetics leads to neurotoxic effects in animal models ranging from cell death to learning and memory deficits. The present study reveals that subtle, albeit long-term changes observed in an adult animal’s learning and memory, after neonatal anesthetic exposure, may likely involve perturbation of specific gene expression patterns. Overall, the research is interesting, well-planned, and has significant results. The paper is accepted after minor corrections as suggested in the comments.

Minor comments

1.      Citation writing format is wrong throughout the paper. Correct it accordingly in IJMS format.

2.      Format of reference writing is wrong. Correct it accordingly in IJMS format.

3.      In Figure 3, you have mentioned Figure 1 in the legend. Correct it.

4.      In Figure 8, the presynaptic neuron (bottom)……….. transcriptionally regulated by Erg1” is unclear. Please slightly improve the first step of the figure (presynaptic neuron).

5.      In Figure 9, in line 473 “pre-synaptic neuron (top)” is mentioned. Is it correct? Or it should be postsynaptic?

6.      In Figure 9, the top part is not looking good either increase the size of the nucleus to cover the maximum area of it or reduce the size. Improve accordingly.

Author Response

We thank the reviewers for their constructive comments. Most issues raised were fair and those have now been addressed below on a point-to-point basis.

Reviewer 1

Minor comments

  1. Citation writing format is wrong throughout the paper. Correct it accordingly in IJMS format
  2. Format of reference writing is wrong. Correct it accordingly in IJMS format.

Thank you for noticing this, we have corrected them to align them with the “Multidisciplinary Digital Publishing Institute” style on Mendeley.

  1. In Figure 3, you have mentioned Figure 1 in the legend. Correct it.

The legend in Figure 3 says:

“Figure 3: Intraperitoneal DEX administration diminished respiratory rate but not oxygen satu-ration. (A) Breaths monitored over time after DEX or saline administration at 0, 10, 20 and 30 minutes after the injection. Bars indicate ± SEM. n = 39-40 animals per condition. (B) Oxygen sat-uration monitored over time after DEX or saline administration at 0, 30 and 60 minutes after the injection. Bars indicate ± SEM. n = 39-40 animals per condition.”

There is no mention of Figure 1 in the legend.

Figure 3 is only mentioned starting in section 2.5, line 263. Similarly, Fig 1 is only mentioned in sections 2.1 and 2.2. We do not understand what the issue raised by the reviewer is. Please, clarify.

  1. In Figure 8, the presynaptic neuron (bottom)……….. transcriptionally regulated by Erg1” is unclear. Please slightly improve the first step of the figure (presynaptic neuron).

We have now modified the sentence on Figure 8 legend to state: “This transmitter exocytosis is mediated by Vamp2, which in turn is transcriptionally regulated by Erg1.”

  1. In Figure 9, in line 473 “pre-synaptic neuron (top)” is mentioned. Is it correct? Or it should be postsynaptic?

The top panel depicts the post-synaptic neuron as it contains PSD95 (Postsynaptic Density Protein 95). Thanks for noticing the typo. Line 478 has been fixed to state POST instead of pre.

  1. In Figure 9, the top part is not looking good either increase the size of the nucleus to cover the maximum area of it or reduce the size. Improve accordingly.

Please note that this is a only drawing simulating the sizes of various intracellular components of cells and is not drawn to scale. The size of the nucleus in a eukaryotic cell should be approximately 10% of the total cell surface/volume. The only reason that it appears larger in Figure 8 in the post-synaptic neuron is to make accommodations so that various transcription factors could be displayed within.

Reviewer 2 Report

This study sought to assess the effects of sevoflurane on learning and memory. Results demonstrated that sevoflurane exposure results in memory deficits in rats. Dexmedetomidine was able to prevent sevoflurane-induced anxiety. Differential gene expression levels were identified after exposure to both sevoflurane and dexmedetomidine, including genes linked to synaptogenesis, neurogenesis, apoptosis, and learning and memory.

Overall, the study was very well-written and carried out, and the figures detailed and comprehensive. I have but a few suggestions on how to better contextualize this clinically relevant study.

Abstract

The reader may be confused by the rapid switch in discussion from learning and memory to anxiety. It would make most sense to first present all the effects of sevoflurane, including anxiety, before introducing the anxiety-relieving effects of dexmedetomidine. Any more detailed flow/presentation can work for the abstract as such.

It would also help to explain that clinically relevant concentrations were used. This will be key for ease of interpretation / clinical translation by clinicians.

It would also help to name how many genes of the 770 were found differentially expressed in the abstract, and perhaps even categorize them (e.g. one quarter linked to XX, one fifth linked to YY, etc.).

Introduction

There is slight inconsistency in your study vision. In the end of the Intro you mention “Secondly, we sought to identify genes that could potentially be involved in the induced long-term learning and memory deficits,” but the reader may also seem to understand that you looked for genes linked to anxiety (post-DEX exposure), since this is also previously mentioned. Do clarify.

Results

Figure 8, 9 are helpful but it would be great if possible to specify them more, e.g. by quantifying the different categories as percentages, or by listing the genes identified under each category. This would help readers versed in genetics to glean information more quickly, with an integral overview.

It would also be of interest to perform a spatiotemporal analysis of such genes in greater detail, or at least discuss this in greater detail: where are they mostly expressed, when, and can any interesting correlations be identified to behavior with regard to a disproportionately important time window/area of the brain (e.g. using Allen Brain Atlas spatiotemporal gene expression data)? If so, clinically this could mean that it would be important simply to avoid such a timing, or use precise genetic tools to minimize negative outcomes from sevoflurane exposure.

Discussion

It would also be interesting for context in a few sentences to briefly discuss/compare sevoflurane and dexmedetomidine to other alternative options, mostly of direct clinical relevance.

Finally, you discuss many single gene effects. Would it be possible to discuss the effects of gene networks/combinations/modules instead, since these are likely more functionally salient than single genes, isolated from their partners mechanistically?

Author Response

We thank the reviewers for their constructive comments. Most issues raised were fair and those have now been addressed below on a point-to-point basis.

Reviewer 3 Report

The authors performed extensive work and presented important results. Yet, some revisions are required on the manuscript to fit for publication as follows:

1. Abstract needs to be rewritten in a more representative and clear way. The background should be more concise. The experimental design should be clarified, as well as the estimated parameter. 

2. Many paragraphs begin with “studies” but end with one reference (E.g. lines 39, 41). Please revise.

3. The presentation of results needs extensive revision as follows:

- All paragraphs describing the importance of the parameter or repletion of the method used should be transferred to the discussion section (E.g. lines 95-101, 125-130,…etc).

- Remove all means values, SEM, and n. It is enough to mention the exact p-value of significance.

- In all figure legends, clarify n=?.

- Why the data in Figure 7 has been presented as means, not means±SEM?

4. The discussion is lengthy with repletion of the introduction section (E.g. lines 498-503). The discussion needs to be refined by directly clarifying the possible interpretations of the results.

5. Material and methods:

- Line 675: on what basis have the authors chosen the tested concentration of dexmedetomidine? Justify with references.

- Line 684: Give the reference to the sevoflurane exposure protocol.

- Line 690: replace sacrificed with euthanized.

- Statistical analysis: Does data meet the assumption of homogeneity of variances and normal distribution? Clarify if the authors run a homogeneity or normality test.

Some English editing is required for grammatical and formatting errors, particularly for letter capitalization.

Author Response

We thank the reviewers for their constructive comments. Most issues raised were fair and those have now been addressed below on a point-to-point basis.

Reviewer 3

  1. Abstract needs to be rewritten in a more representative and clear way. The background should be more concise. The experimental design should be clarified, as well as the estimated parameter. 

We thank the referee for their insightful comments. This paper is now revised in light of all three referee's comments. The issues raised by this referee are addressed below on a point-to-point basis.

Unfortunately, the limited space constraints force us to provide an abbreviated version of the Abstract where the rationale, experimental design, results and future perspective had to be crammed into the allocated space. However, all salient aspects of the study have been described properly, and as written, it will force the reader to ascribe to the entire paper rather than scanning through the abstract alone. We have therefore adhered to the abstract word limit and have described as much as we could in 245 words (we are already over the 200 permitted words limit).

  1. Many paragraphs begin with “studies” but end with one reference (E.g. lines 39, 41). Please revise.

Once again, to avoid citing too many references, we have opted to cite comprehensive review articles which cover all aspects of the research that we have described in this paper. Thus, Line 39 refers to a review paper where all clinically relevant anesthetics had been described and discussed in detail [3]. There is no reference on line 41 and if you are referring to reference 8 on line 43, it is an FDA statement [4]

  1. Iqbal, F.; Thompson, A.J.; Riaz, S.; Pehar, M.; Rice, T.; Syed, N.I. Anesthetics: From Modes of Action to Unconsciousness and Neurotoxicity. J. Neurophysiol. 2019, 122, 760–787, doi:10.1152/jn.00210.2019.
  2. FDA Drug Safety Communication: FDA Approves Label Changes for Use of General Anesthetic and Sedation Drugs in Young Children | FDA Available online: https://www.fda.gov/drugs/drug-safety-and-availability/fda-drug-safety-communication-fda-approves-label-changes-use-general-anesthetic-and-sedation-drugs (accessed on 13 October 2021).
  3. The presentation of results needs extensive revision as follows:
  4. All paragraphs describing the importance of the parameter or repletion of the method used should be transferred to the discussion section (E.g. lines 95-101, 125-130,…etc).

We see the referee's point but beg to differ. In order for the reader to understand the context and the rationale of any particular experiment that we performed, it is imperative that a brief context be provided for comparative purposes and to better help understand the figures without invoking extensive discussion. We feel that in such a study, a brief layout of the protocol is helpful rather than a distraction.

  1. Remove all means values, SEM, and n. It is enough to mention the exact p-value of significance.

Once again, we beg to disagree as these mean values and SEM provide a more direct and accurate depiction of the data as stated in the text.

  1. In all figure legends, clarify n=?.

We thank the referee, this has now been added accordingly.

  1. Why the data in Figure 7 has been presented as means, not means±SEM?

Figure 7 is describing the fold change in the gene expression. There is no SEM for these values as the program will only give fold changes, therefore mean and SEM cannot be plotted in this type of analysis

  1. The discussion is lengthy with repletion of the introduction section (E.g. lines 498-503). The discussion needs to be refined by directly clarifying the possible interpretations of the results.

Please note that there are considerable discrepancies in the literature regarding both of these anesthetics, therefore it is important to provide a more detailed and comparative account in the discussion. In the introduction, however, we merely alluded to these studies to set the context of our paper. We strongly feel that orientating the readers towards the most recently published studies in the literature helps them recognize gaps in the field vis-à-vis the lack of genetic analysis that we have achieved in this study.

  1. Material and methods:
  2. a) Line 675: on what basis have the authors chosen the tested concentration of dexmedetomidine? Justify with references.

The tested concentrations conform to clinically relevant data as stated in line 676, please see the reference.[5]

  1. Endesfelder, S.; Makki, H.; Haefen, C. von; Spies, C.D.; Bührer, C.; Sifringer, M. Neuroprotective Effects of Dexmedetomidine against Hyperoxia-Induced Injury in the Developing Rat Brain. PLoS One 2017, 12, e0171498, doi:10.1371/JOURNAL.PONE.0171498.
  2. b) Line 684: Give the reference to the sevoflurane exposure protocol.

Our previously published paper describing sevoflurane exposure has now been added.

  1. c) Line 690: replace sacrificed with euthanized.

As per the referee’s suggestion, the sentence has now been modified.

  1. d) Statistical analysis: Does data meet the assumption of homogeneity of variances and normal distribution? Clarify if the authors run a homogeneity or normality test.

Thank you for noting this and we regret this oversight. We did both One-way Anova Dunnet post-hoc (multiple comparisons among all groups) and Two-way Anova Tukey’s post hoc comparisons and these have now been corrected.

Comments on the Quality of English Language

Some English editing is required for grammatical and formatting errors, particularly for letter capitalization.

We have gone over the paper and fixed sentences where we felt that there was a need.

Round 2

Reviewer 3 Report

No further comments are to be addressed

The manuscript has been significantly improved.